# Incorporating Worker Perspectives into MTurk Annotation Practices for NLP

**Olivia Huang**
UC Berkeley

**Eve Fleisig**
UC Berkeley

**Dan Klein**
UC Berkeley

{efleisig, oliviahuang, klein}@berkeley.edu

## Abstract

Current practices regarding data collection for natural language processing on Amazon Mechanical Turk (MTurk) often rely on a combination of studies on data quality and heuristics shared among NLP researchers. However, without considering the perspectives of MTurk workers, these approaches are susceptible to issues regarding workers' rights and poor response quality. We conducted a critical literature review and a survey of MTurk workers aimed at addressing open questions regarding best practices for fair payment, worker privacy, data quality, and considering worker incentives. We found that worker preferences are often at odds with received wisdom among NLP researchers. Surveyed workers preferred reliable, reasonable payments over uncertain, very high payments; reported frequently lying on demographic questions; and expressed frustration at having work rejected with no explanation. We also found that workers view some quality control methods, such as requiring minimum response times or Master's qualifications, as biased and largely ineffective. Based on the survey results, we provide recommendations on how future NLP studies may better account for MTurk workers' experiences in order to respect workers' rights and improve data quality.

## 1 Introduction

Amazon Mechanical Turk (MTurk) is an online survey platform that has become increasingly popular for NLP annotation tasks (Callison-Burch and Dredze, 2010). However, data collection on MTurk also runs the risk of gathering noisy, low-quality responses (Snow et al., 2008) or violating workers' rights to adequate pay, privacy, and overall treatment (Xia et al., 2017; Hara et al., 2018; Kim et al., 2021; Lease et al., 2013; Shmueli et al., 2021). We conducted a critical literature review that identifies key concerns in five areas related to NLP data collection on MTurk and observed MTurk worker opinions on public forums such as Reddit, then developed a survey that asked MTurk workers about open questions in the following areas:

**Task clarity.** How can requesters ensure that workers fully understand a task and complete it accurately? Surveyed workers expressed a desire for more examples and more detailed instructions, and nearly half desired more information about the downstream context in which an annotation will be used. In addition, they indicated that task clarity is a major factor in their judgment of how responsible and reliable a requester is (Section 3).

**Payment.** What is an appropriate level of pay on MTurk? Are there tradeoffs to different amounts or methods of pay? Respondents indicated that there is often a threshold pay below which they will refuse to complete a task, suggesting that pay rates below minimum wage are not only unethical, but also counterproductive to worker recruitment. However, very high pay attracts spammers, which moreover forces attentive workers to complete tasks more quickly before spammers take them all, meaning that response quality likely decreases across all workers. Workers expressed mixed opinions over bonus payments, suggesting that they are most effective if they are added to a reasonable base pay and workers trust the requester to deliver on the bonus. (Section 4).

**Privacy.** What methods best ensure worker privacy on MTurk? How do workers react to perceived privacy violations? Workers report that they often respond to questions for personal information untruthfully if they are concerned about privacy violations. Nearly half of workers reported lying on questions about demographic information, which means that tasks relying on demographic data collection should be very careful about collecting such information on MTurk (if at all). Some of these issues can be partially mitigated if requesters

have good reputations among workers and are clear about the purpose of personal data collection, since spam requesters are common on MTurk. (Section 4).

**Response quality.** What methods best ensure high-quality responses and minimize spam? Requesters must consider how traditional response filters have the side effect of influencing workers' perception of the task, and in turn, their response quality. The rising importance of incorporating worker perspectives is underscored by the establishment of organizations such as Turkopticon, which consolidates workers' reviews of requesters and advocates for MTurk workers' rights, including petitioning to "end the harm that mass rejections cause" (Turkopticon, 2022). We find that workers consider some common quality control methods ineffective and unfair. For example, workers reported that minimum response times often make them simply wait and do other things until they hit the minimum time, while allowing only workers with Masters qualifications to complete a task excludes all workers who joined after 2019, since the qualification is no longer offered. In addition, workers are very concerned about their approval rates because requesters often filter for workers with very high approval rates. Respondents expressed frustration over having work rejected, especially automatically or without justification, and receiving no pay for their labor. Workers thus avoid requesters with low approval rates or poor communication (Section 6).

**Sensitive content.** Tasks in which workers annotate sensitive content can pose a psychological risk, particularly since MTurk workers often face mental health issues (Arditte et al., 2016). Though it is common practice to put general content warnings before such tasks (e.g. "offensive content"), nearly a third of surveyed workers expressed a desire for specific content warnings (e.g., "homophobia") despite this being uncommon in current research practices (Section 7).

Common themes emerge across these topics regarding the importance of maintaining a good reputation as a requester, understanding worker incentives, and communicating clearly with workers. In Section 8, we discuss the implications of worker preferences for survey design and provide recommendations for future data collection on MTurk.

## 2 Survey Design

We conducted a literature review and examined posts on r/mturk, a Reddit forum that MTurk workers use to discuss the platform, to identify the areas of uncertainty discussed in sections 3-7. Then, to collect data on MTurk workers' experiences, we conducted a Qualtrics survey posted as a task on MTurk. Our survey was broadly subdivided into sections on payment, sensitive questions, response quality, and miscellaneous issues (mainly task clarity and context). As a preliminary filter for response quality, we required that all respondents have a 97% HIT approval rate, at least 100 HITs completed, minimum of 18 years of age, and English fluency. This is a lower HIT approval rate than is typically used, which can increase the rate of spam; however, we aimed to collect opinions of workers who might be excluded by high approval rate filters, and therefore manually reviewed the data afterwards to remove spam responses by examining responses to the required free text fields. We collected 207 responses from our survey over one week, of which 59 responses were dropped for spam or incompleteness (we discuss further implications of high spam rates in Section 8). Each annotator was paid $2.50 to complete the survey, based on the estimated completion time and a minimum wage of $15/hour. Appendices A and B contain details on informed consent, the full text of the survey, and data cleaning details.[1]

The following sections address each area of concern raised in Section 1, discussing previous research on MTurk annotation practices, the open questions raised by the literature, the survey questions we included on those topics, and the results and implications of the survey responses.

## 3 Task Clarity

On MTurk, *requesters* publish "human intelligence tasks" (or *HITs*) for MTurk workers to complete in exchange for a monetary incentive. Best practices for question phrasing and ordering are consistent with overall survey design guidelines, which encourage including clear instructions for how to use the platform, clearly outlining requirements for acceptance, including label definitions and examples, and avoiding ambiguous language (Gideon, 2012). To look for strategies for minimizing confusion in survey questions, our survey asked what additional

---

[1]This study underwent IRB review and annotators provided informed consent prior to participation.

provided information, such as context and purpose of the task, would be most helpful for reducing ambiguity or confusion.

To ensure that results are accurate and useful, it is important to carefully conceptualize the task (define what quality is being measured by the study) and operationalize it (decide how the study will measure the construct defined by the conceptualization) along with clear compensation details (discussed further in Section 4 and Jacobs and Wallach, 2021). In a Reddit thread on r/mturk, worker u/dgrochester55 (2022) stated that "we are not likely to take your survey seriously if you as a requester aren't taking your own survey seriously" due to issues such as "poor pay, irrelevant subject" or being "badly put together." Thus, task clarity not only ensures that the worker can provide accurate answers, but also serves to increase their effort levels and willingness to complete the task itself.

Our survey sought to further clarify the importance of task clarity by asking whether it influences effort level and willingness to start a task. In addition, we asked a free response question about common areas of confusion to observe common pitfalls in designing MTurk tasks.

### 3.1 Context

One way to improve task clarity is to provide additional context for the task. Context is commonly provided in two ways. The first way is to provide additional words, sentences, or paragraphs in the annotated text itself. Providing context can increase disagreement among responses, and can be useful if the goal is to mimic the distribution of differing human judgments (Pavlick and Kwiatkowski, 2019). The second way is to introduce background information about the speaker, such as age, gender, race, socioeconomic status, etc. In tasks such as annotating for the presence of harmful language, such information can affect an annotator's perception of text as influenced by individual biases.

Sometimes, context is inherent in the language itself, instead of explicitly provided. For example, some annotators may incorrectly interpret African American English (AAE) as offensive (Sap et al., 2019). Other forms of context include grammaticality, location, and relative positions of power between the speaker and the audience.

Thus, it is important to note that even without the presence of additional context, text is inherently contextual. It is important to identify and control

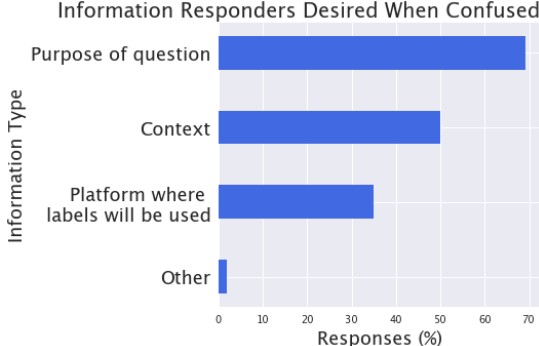

Figure 1: Types of information that respondents felt would have helped them answer questions they found confusing.

for such qualities in annotation tasks. To better understand how context can influence annotation tasks, we asked MTurk workers what additional information is most useful to them when given a question with an unclear answer.

### 3.2 Survey Results on Task Clarity

We found that task clarity and fair payment are key factors that determine both MTurk workers' willingness to start a task and the amount of effort they put into it. In our survey, 26.3% of respondents stated that difficulty of understanding the task influences whether they start the task, and 31.6% indicated that it influences the amount of effort they put into the task. One respondent suggested providing additional, more comprehensive examples, since requesters "often skimp on these or only provide super basic, obvious examples." When respondents were asked to rank the importance of different qualities in determining how "reasonable" and "reliable" the requester is (from one to six), clarity of instructions was ranked second highest with an average rank of 2.71 (payment was ranked highest; see Section 4). These results reinforce the importance of task clarity in improving workers' perceptions of the task and its requester.

Workers often reported being confused by annotation tasks that involve sentiment analysis and subjective ratings. For such questions, the most commonly preferred additional information was the purpose of the question (69.1%), context (50.0%), and the platform the annotations would be used for (34.9%). This suggests that providing an explanation for the downstream use case of the task significantly aids workers in understanding tasks and providing quality annotations.

## 4 Payment

MTurk's platform, which enables researchers to collect large amount of data at low cost, does not currently regulate compensation beyond a \$0.01 minimum per task that disregards task completion time. The mean and median hourly wages for MTurk workers are \$3.13 and \$1.77 per hour, respectively. 95% of MTurk workers earn below minimum wage requirements in their geographic location (Hara et al., 2018), and improvements in household income for U.S. MTurk workers lag behind the general U.S. population (Jacques and Kristensson, 2019). Low payment is not only unfair to workers, but can also serve as an additional sign to workers of an unreliable requester to avoid due to high potential of having a HIT rejected. u/Bermin299 (2022) claims that "Requestors that pays cheaply are some of the most trigger happy people when it comes to handing out rejections."

Most researchers agree on using minimum wage as a baseline hourly rate for MTurk work, though exact recommendations vary due to differences in the minimum wage worldwide (Hara et al., 2018). A common sentiment among MTurk workers is to pay at a rate of \$15 per hour (u/Sharpsilverz, 2022). Whiting et al. (2019) found that workers are likely to overestimate their work time by 38% of the observed task time. Thus, the amount of time used to determine minimum wage can roughly account for this overestimation if using workers' self-reported work times as a metric. Previous research is divided on whether extra pay beyond minimum wage improves response quality. Callison-Burch and Dredze (2010) gave anecdotal evidence that unusually high payments, such as \$1 for a very short task, may encourage cheating.

However, Snow et al. (2008) found that non-guaranteed payments, which are paid only after submitted work is approved for work quality, are sometimes effective at improving response quality. MTurk's bonus payments can also be non-guaranteed payments, and they are given to workers after they have completed the task in addition to the advertised payment rate.

Given the wide range of payment options and formats, along with varying guidelines on the effectiveness of different payment options, we include multiple questions in our survey regarding MTurk workers' perception and response to payments. These included workers' perception of normal and bonus payments, how bonus versus normal payments affect effort, how payment influences likelihood of starting or completing the task, and how payment affects the degree to which requesters are perceived as "reasonable" and "reliable." We also asked MTurk workers about their MTurk income relative to the opportunity cost of their work, as well as their own thresholds for the minimum pay rate at which they would do a task.

### 4.1 Survey Results on Payment

Survey results indicate reasonable payment to be around minimum wage; significantly higher or lower payment appears to be detrimental to the quality of responses. Respondents' minimum pay rate at which they would be willing to complete a task was \$13.72 per hour on average (median of \$12 per hour). 64.5% of respondents stated that the rationale behind their threshold is that they want to earn a wage on MTurk comparable to what they would make elsewhere. These numbers are significantly above the current mean and median, though still below minimum wage. Despite this, the survey surprisingly indicates that 70.4% of respondents make more on MTurk than they would make elsewhere, and 22.4% make the same amount. However, payment should not be set too high, as one worker stated that "[e]xtremely high pay rates cause rushing in order to complete as many as possible prior to the batch getting wiped out by other workers."

Workers responded that both higher normal payments and bonus payments increase the amount of effort they put into a task. 73.0% of respondents indicated that bonus payments will increase their effort levels while 49.3% selected high(er) payments. However, some respondents noted that bonus payments are only a significant incentive if they are from a reputable requester. Uncertainty about whether the bonus will be paid is the main concern, with one worker noting that "regular payments are guaranteed within 30 days, whereas bonuses can take however long the req decides to take to pay you (assuming they even do)." Thus, the effectiveness of bonus payments varies with workers' perception of the requester's reliability and reputation.

## 5 Privacy

Demographic questions are commonly used in surveys to collect respondents' information. However, some MTurk workers may view these questions as

threats to their privacy and decide not to complete the survey. To clarify which questions are more commonly viewed as "sensitive," our survey asked MTurk workers to indicate types of questions, including demographic questions and questions about offensive topics or mental health, that cause them to not complete a task or answer untruthfully. We also asked for reasons would cause them to answer a question untruthfully, including whether they believe that the question is a privacy violation or that it will be used against them.

It is especially important to build trust and the perception of fairness between the requester and the annotator for surveys that ask more personal questions. Xia et al. (2017) found that crowdworkers' major areas of concern regarding data privacy involved collection of sensitive data, information processing, information dissemination, invasion into private lives, and deceptive practices. To minimize these concerns, they proposed providing information about the requester, being specific about why any private information is collected and how it will be used, not sharing any personal information with third parties, and avoiding using provided emails for spam content outside the context of the survey. Turkopticon's Guidelines for Academic Requesters recommend that requesters provide information that clearly identifies who the requester are and how to communicate with them, which indicates that the requester is "accountable and responsible", as well as avoiding collection of workers' personally identifying information (Turkopticon, 2014). Building on this, our survey asked what additional information, such as content warnings and mental health resources, would be most helpful in surveys with sensitive questions.

Best practices on demographic questions include ensuring inclusivity for questions on gender identity by allowing for open ended responses and considering that terminology changes over time (e.g. "How do you currently describe your gender identity?"), distinguishing between ethnicity and race, and permitting annotators to indicate multiple responses for race, ethnicity, and gender in case they identify with multiple categories (Hughes et al., 2016). In our survey, we asked MTurk workers if they encountered questions where the responses provided did not adequately capture their identity.

Another MTurk worker vulnerability is the use of Worker IDs. Each MTurk worker has an identifying Worker ID attached to their account, associated

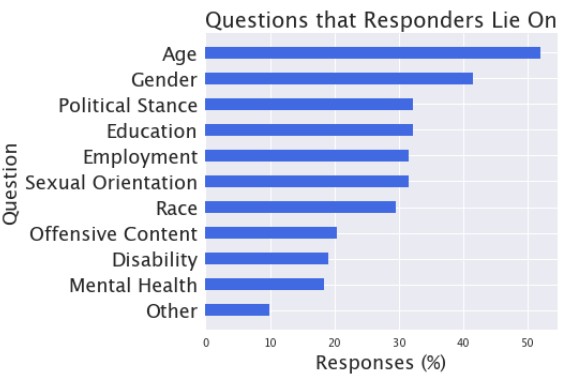

Figure 2: Types of personal questions that workers report will cause them to answer untruthfully. Over half of annotators report that questions about age will make them answer untruthfully, and over 40% report the same about gender.

with all of their activities on Amazon's platform, including responses to MTurk tasks and activity on Amazon.com (Lease et al., 2013). As a result, any requester with access to a Worker ID (which is automatically included with survey responses) can identify the associated MTurk worker's personal information through a simple online search. This can return private information such as photographs, full names, and product reviews (Lease et al., 2013). As this is a vulnerability in MTurk's system that cannot be removed by the requester, it is common for requesters to simply drop the Worker IDs from the dataset before continuing with any data analysis. In our survey, we ask MTurk workers whether they are aware of the risk of their Worker IDs being attached to MTurk responses and if that knowledge is a concern that influences their behavior.

## 5.1 Survey Results on Privacy

Our survey results indicate that concerns about privacy often lead to untruthful answers. Overall, 79.6% of respondents stated they will answer a question untruthfully if they feel that it is a privacy violation, and 17.1% stated they will do so if they have concerns that the questions will be used against them. In regards to Worker IDs, 51.3% of workers stated that they are aware that they exist, but it has not caused them to change their behavior.

In addition, workers often feel that demographic questions do not capture their identity. This is most often an issue for questions about gender (38.2%), age (38.2%), and sexual orientation (37.5%). Workers also frequently answer demographic questions untruthfully (Figure 2), especially those regarding age (52.0%), gender

(41.5%), and education (32.2%). 9.87% of respondents indicated that other questions caused them to answer untruthfully, and most frequently mentioned questions requesting a phone number, name, or zipcode. Other respondents were concerned that their education level could be held against them.

## 6 Response Quality

Response quality can be maximized by ensuring that the task is clear and well-designed through a preliminary pilot task (annotation tasks released before the actual task to fine-tune the survey) and filtering respondents to include those who are more likely to honestly complete the task (Gideon, 2012). Requesters can also work towards maximizing response quality by using various functionalities provided by MTurk's filtering software.

### 6.1 MTurk Qualifications

MTurk provides several mechanisms for improving response quality. These include Masters Qualifications (which filter for "Masters Workers", selected on the basis of successful completion of a wide range of MTurk tasks over time), System Qualifications (e.g. HIT approval rate, location), and Premium Qualifications (e.g., income, job function). MTurk also has "Qualifications Types you have created," which allow requesters to assign specific workers with a custom score between 0-100 to determine their eligibility for tasks. One challenge to filtering MTurk workers is balancing the use of filters to minimize spam responses while still allowing enough real workers to respond in order to collect a large enough dataset. Filtering on location is common because it helps to filter out MTurk workers who may be unqualified to complete tasks that require proficiency in certain languages or experience with a certain culture or environment (Karpinska et al., 2021).

To prevent overfiltering respondents at the beginning, some papers suggest keeping numerical thresholds relatively low. Peer et al. (2014) stated that high-reputation and high-productivity workers can be filtered using 500 approved HITs and 95% HIT approval rate, while a looser lower bound to avoid low-reputation and low-productivity workers is 90% approval rate and 100 approved HITs. However, more recent research has found that around 2500 approved HITs and 99% HIT approval rate is a more effective filter for MTurk surveys, since approval rates below that result in a significant drop

in response quality (Kummerfeld, 2021). Meanwhile, researchers such as Ashida and Komachi (2022) used a 98% HIT approval rate to allow more workers to attempt HITs. This aligns with views expressed by MTurk workers that a 99% approval rate is unreasonably difficult to obtain, as it requires a near perfect record when HITs are often rejected without a logical reason or explanation (u/Lushangdewww, 2019). Furthermore, a 99% HIT approval rate largely serves to unfairly discriminate against newer workers (rather than low quality workers), due to the relative margin of error based on the total number of HITs completed (u/ptethesen, 2022). Turkopticon encourages providing clear conditions for rejection due to the permanent mark that rejection leaves on a worker's record (Turkopticon, 2014).

Many MTurk requesters also use time as a means of filtering responses both as workers are completing tasks (e.g. through timers) and after receiving the complete dataset, by removing the top and bottom percentile of response times (Justo et al., 2017). In response to one such task warning that "speeders" would be rejected, however, an r/mturk user stated that they "did that survey, made some lunch, checked my text messages then submitted" (u/gturker, 2022). Such forum posts suggest that these filters encourage workers to work on other tasks to pad time rather than spend more time thinking about their answers.

While several papers outline strategies to ensure high response quality, there is little work on how such measures are received by MTurk workers themselves. Thus, our survey asked MTurk workers about how strategies such as limiting response time ranges and adding time limits affect their response quality. In addition, we asked if any of these measures are perceived as unreasonable. Lastly, we asked MTurk workers to provide their own input on reasonable checks for response quality and whether any existing qualifications are perceived as unfair or unreasonable.

### 6.2 Recent LLM Developments and Concerns

The rise of large language models (LLMs) poses a significant risk to response integrity. Veselovsky et al. (2023) found that 33 to 46% of workers used LLMs to complete a text summarization task on MTurk. Furthermore, these responses may be difficult to detect, as LLMs such as ChatGPT outperform humans on some annotation tasks (Gilardi

et al., 2023). In our survey, we found that required, open-ended text responses made it easiest to spot spam respondents who might pass unnoticed in rating or classification tasks, but such manual methods may become less effective in the face of better text generation tools. However, multi-stage filtering procedures, such as Zhang et al. (2022)'s pipeline that uses a qualification task and a longer endurance task to identify a high-quality worker list before the main task begins, can help to identify a trustworthy pool of workers to minimize spam responses. AI-generated text detectors may also help to flag potential spam responses in these filtering pipelines (Veselovsky et al., 2023).

### 6.3 Survey Results on Response Quality

One common metric to filter for response quality is response time. Overall, respondents indicated that task timers and time estimates are well-calibrated. However, we also found that time-based filters are largely unhelpful and counterproductive. When given minimum required response times, $41.5\%$ of workers reported that they spend the additional time working on other things to extend their completion time. $22.4\%$ of survey respondents reported having had a HIT unfairly rejected due to responding too fast. Meanwhile, time limits can reduce response quantity and quality. One respondent explained that "it is frustrating when you have put in a great deal of conscientious effort only to find the time has expired and you cannot submit a HIT." These results align closely with the comments made in the Reddit thread described in the previous section (u/LaughingAllTheWay83, 2022a).

The filter most commonly seen as unreasonable is the Masters qualification ($55.9\%$), because MTurk has reportedly stopped giving it out. One respondent explained that "Masters is a dead qual. Hasnt been issued to anyone since 2019. It is the single biggest gatekeeper to success for newer turkers like myself." Thus, MTurk requesters who require a Masters qualification unknowingly filter out all workers that joined after this time.

On average, respondents indicated that a reasonable HIT approval rate is $93.6\%$, while the median response was $98\%$. Overall, MTurk workers reported a strong incentive to maintain a high HIT approval rate, as $40.1\%$ of survey respondents stated that the approval rate of the requester influences the amount of effort they put into their responses and $36.2\%$ state that it influences whether they even start a task. Thus, a good practice for rejecting HITs is is to provide a clear rationale for why a HIT is being rejected as feedback for future work.

## 7 Sensitive Content

MTurk surveys involving more sensitive NLP tasks can pose psychological risks. MTurk workers are more likely to have mental health issues than the general population, making them a vulnerable population (Arditte et al., 2016). NLP tasks that pose a potential risk to workers include labeling hate speech applicable to the worker's own background, identifying adult or graphic content, or questions about workers' mental health. There is usually higher risk with tasks that involve language that provides strong emotional stimuli (e.g. offensive tweets), which can potentially cause trauma (Shmueli et al., 2021). In addition, work may be more personally damaging when the offensive content directly applies to the worker (Sap et al., 2020). It may be beneficial to modify studies being completed by targeted demographic groups with additional safeguards and warnings in the survey regarding potential harm or offense.

In addition to exposing workers to potential psychological harm, the presence and framing of sensitive questions can cause workers to leave tasks incomplete or answer untruthfully. u/LaughingAllTheWay83 (2022b) described providing "accurate and thoughtful answers 99% of the time, the exception being in regards to my own mental health...Requesters have been known to overstep their lane and contact the police for a welfare check and I'm not opening myself up to that possibility for an extra $2, ya know?" Thus, it is vital for researchers to identify potentially sensitive content and take steps to protect workers from possible harm.

Generally, researchers are aware of the presence of offensive content, as reflected in content warnings included in the beginning of published papers (e.g., Sap et al., 2020). However, there is a clear need for standardized best practices due to the highly vulnerable nature of such MTurk work.

A relevant example is Facebook's partnership with Sama, a company that pays workers to identify illegal or banned content on social media (Perrigo, 2022). An interview with the workers revealed high levels of psychological distress due to insufficient breaks and subpar in-person resources. In an online setting such as MTurk, such personal-

ized care is even less accessible, making it more difficult to provide support for vulnerable workers. These workers' experiences exemplify the difficulties faced by the large, invisible workforce behind MTurk and powerful NLP applications, also described by Gray and Suri (2019) in their discussion of the typically unregulated "ghost work" that powers machine learning.

Common safeguards include listing clear descriptions of the benefits and risks of the task, providing mental health resources for workers, using data safety monitoring boards, adhering to institutional review board policies, and following confidential data code and informed consent procedures (Kim et al., 2021). We look to further add to this list of best practices by asking survey respondents whether there are questions that will immediately deter them from completing a task or to answer untruthfully and preferred actions or resources provided by requesters to minimize possible harm.

## 7.1 Survey Results on Sensitive Content

Sensitive survey questions, such as questions that ask workers to annotate harmful or offensive content (e.g. hate speech), not only expose workers to potential harm, but also often lead to inaccurate or incomplete responses. When we asked for specific types of questions that cause workers to decide to not complete a task, 23.7% of respondents indicated offensive content, and 19.1% indicated questions concerning mental health. As seen in Figure 2, 20.4% of workers reported answering questions on offensive questions untruthfully and 18.4% did so for questions regarding mental health.

In a survey that contains potentially sensitive questions, 43.4% of respondents most preferred general content warnings (whether the survey contains sensitive content), 30.9% preferred specific content warnings (the type of sensitive content the survey contains), and 23.7% preferred mental health resources (Figure 3). These numbers indicate that a far higher number of workers prefer specific content warnings than there are currently in MTurk studies.

## 8 Recommendations

We consolidate our findings on worker perspectives to arrive at several recommendations.

**Pay at least minimum wage, but not exceptionally higher.** Payment can influence MTurk workers' perception of the task as well as their effort

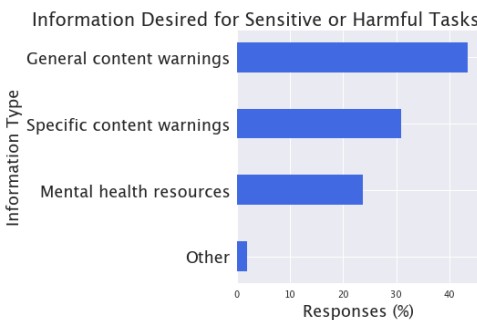

Figure 3: Information desired for tasks about sensitive or harmful content. Nearly a third of annotators desire specific content warnings (e.g., "homophobia"), though general content warnings (e.g., "offensive content") are more common in current tasks.

levels. We suggest paying around the minimum wage of $15 per hour (slightly above the median minimum threshold of $12, and in accordance with the views expressed by r/MTurk users), but not too much higher, which decreases response quality.[2] Additional budget may be used as bonus payments, which are effective incentives when provided by reputable accounts.

**Minimize the collection of private information, and be transparent.** Workers are highly sensitive to questions that they see as violations of privacy, such as demographic questions. It is thus prudent to minimize their use, provide a brief explanation for their importance, clarify that the answers are confidential and will not be held against the respondent, and include an option for respondents not to answer. To protect worker privacy, always drop Worker IDs before working with the dataset. Also follow worker-centered policies such as Turkopticon's Guidelines for Academic Requesters (Turkopticon, 2014) to maintain transparent requester policies and communication.

**Avoid time-based filters and the Masters qualification.** Our results indicate that both lower and upper bounds on completion time are flawed methods of quality filtering. Thus, we discourage rejecting HITs based on time, barring humanly impossible extremes. In addition, to avoid unnecessarily filtering out workers who joined MTurk later, do not use the outdated Masters' Qualification. For high-quality responses, a 98% HIT approval rate filter adheres to both worker preferences and re-

---

[2]We note that living wages vary by region and inflation, so the ideal pay is likely to change over time and may depend on the region from which MTurk workers are recruited.

search findings, as it exactly equals the median value proposed in our survey results. Clearly outline the requirements for task approval to increase trust between the worker and the requester.

**Consider effects on worker approval rate when controlling for quality.** Since MTurk spam rates are high (as we saw firsthand), quality filtering is key to obtaining good results. However, workers are sensitive to the fact that drops in approval rate bar them from future tasks, and so are both less willing to accept tasks from requesters with lower approval rates, and frustrated when work is rejected without explanation. As a result, it is best to provide explanations when rejecting workers. Paid qualification tasks that filter and/or train workers before they begin a larger main task, such as the pipeline proposed by Zhang et al. (2022), can help to minimize spam responses without needing to reject many responses on the main task. As text generation tools become more powerful, AI-generated text detectors may also help to flag potential spam responses (Veselovsky et al., 2023).

**Provide context for subjective tasks.** When possible, preface questions with an explanation of the purpose of the annotation task, such as the downstream use case. Additional context is also helpful, such as whether the text is from a longer passage or was sourced from a specific platform.

**Provide general and specific content warnings.** If questions in sensitive areas such as offensive content or mental health are necessary, provide general and specific warnings for sensitive content, an explanation of the questions' purpose, and an overview of how the data will be used.

## Limitations

We surveyed MTurk workers from the United States; the views of these workers may differ from those of workers in other parts of the world. In addition, because the survey was completed by the first 207 workers to see the posted task, there may be sampling bias in the workers who answered (i.e., ones who go on MTurk more often or are quicker to answer tasks are more likely to have filled out the survey). Future studies with a larger sample pool of workers from different parts of the world, and on different crowdworking platforms, could help to examine the extent to which these results generalize.

## Ethical Considerations

Understanding the factors that make MTurk workers more likely to trust a requester and thus more likely to provide personal information or accept tasks with unusual payment schemes (e.g., low normal pay and high bonus pay) could be misused by spam requesters or phishers to maliciously acquire more information from MTurk users or extract their labor without fair pay. We understand that these are potential consequences of our research, but hope that they are outweighted by the potential benefits of good-faith requesters understanding how to improve the effects of their tasks on workers.

## Acknowledgments

Many thanks to Nicholas Tomlin and Eric Wallace for their feedback on earlier drafts of this paper.

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

# A Informed Consent

## A.1 Key Information

You are being invited to participate in the online version of our research study on Amazon Mechanical Turk (MTurk). Participation in research is completely voluntary. The purpose of the study is to examine the best practices and common pain points involved in conducting annotation tasks on MTurk. The study will take approximately 10 minutes, and you will be asked to respond to questions regarding your experience and opinion on conducting annotation tasks. Risks and/or discomforts may include potentially sensitive questions that may cause discomfort in the survey respondent. There is no direct benefit to you. The results from the study may help improve general MTurk workers' experiences in future MTurk annotation tasks.

**Introduction and Purpose**

(Researcher names) We would like to invite you to take part in our research study, which concerns observing MTurk workers' experiences on the platform in order to improve understanding on best practices and common pain points involved in conducting annotation tasks. The key research areas involve 1) fair payment methods and its effectiveness in incentivizing workers, 2) administration of sensitive questions, 3) methods for maximizing response quality, and 4) miscellaneous questions on survey design and MTurk workers' experiences.

**Procedures**

If you agree to participate in this research, we will ask you to complete the attached Qualtrics survey included in the MTurk task. The survey will involve questions about payment, administration of sensitive questions, maximizing response quality, survey design, and MTurk workers' experiences, and should take about 10 minutes to complete.

**Benefits**

There is no direct benefit to you from taking part in this study. It is hoped that the research will increase knowledge of the risks involved in MTurk surveys. For example, participants may learn about the fact that Worker IDs are shared to survey requesters. In addition, the purpose of the study itself is to improve the MTurk survey experience for MTurk workers by making it safer, less invasive, and more reasonable. Thus, the MTurk workers completing the task will benefit from the survey findings.

**Risks/Discomforts**

Although the survey does not explicitly ask any demographic questions, it asks about participants' perceptions and opinions on sensitive questions such as personal demographic information and offensive content. These questions may be a possible area of discomfort for the participant. As with all research, there is a chance that confidentiality could be compromised; however, we are taking precautions to minimize this risk.

**Confidentiality**

Your study data will be handled as confidentially as possible. If results of this study are published or presented, individual names and other personally identifiable information will not be used. Amazon Mechanical Turk automatically stores a list containing the Worker IDs of all participants who work on a survey requester's task, but these Worker IDs will not be linked to the anonymous Qualtrics responses, will not be downloaded, and will only be used to ensure payment. We will delete the Worker IDs on the Mechanical Turk platform immediately once participants have been paid. We will not obtain any identifiers linked to individual Qualtrics survey responses and there is no other potentially identifiable information collected in the survey. The Qualtrics survey responses will be downloaded immediately after the required number of responses is reached. After they are downloaded, they will be deleted from the Qualtrics platform within one week. The survey data will be secured on two password-protected computers, and it will be retained for no more than 6 months. After this time period, only the aggregate stats will be retained. We will not be transferring any identifiable data after results are downloaded from Amazon Mechanical Turk. When the research is completed, we will save the data for possible use in future research done by us or others. We will retain these records for up to 6 months after the study is over. The same measures described above will be taken to protect confidentiality of this study data. Your personal information may be released if required by law. Authorized

representatives from the following organizations may review your research data for purposes such as monitoring or managing the conduct of this study: University of California. Identifiers might be removed and the de-identified information used for future research without your additional informed consent.

**Compensation**

Participants will be compensated with $2.50 upon completing the survey on Amazon Mechanical Turk.

**Rights**

Participation in research is completely voluntary. You are free to decline to take part in the project. You can decline to answer any questions and are free to stop taking part in the project at any time. Whether or not you choose to participate, to answer any particular question, or continue participating in the project, there will be no penalty to you or loss of benefits to which you are otherwise entitled.

**Questions** [Contact Information]

If you agree to take part in the research, please print a copy of this page to keep for future reference, then check the "Accept" box below.

## B    Survey and Data Cleaning Details

Figures 4 through 10 contain the full text of the survey.

After data was collected, the mandatory free text responses were manually reviewed by two authors for spam, including free text responses that were identical for multiple survey responses, or direct copies of text from the question with nothing added; numbers in response to questions that do not ask for numbers; or responses that were incoherent. We also removed incomplete responses (where the survey was only partially answered).

## MTurk Annotation Practices for NLP Survey Questions

Q1 How does the amount of money you make on Mechanical Turk (MTurk) compare to the amount you would be able to make elsewhere?

○ I make more on MTurk

○ I make the same amount of money

○ I make more elsewhere in the real world

(Read before completing questions 2-4) There are a couple payment features offered on Amazon Mechanical Turk. The relevant ones are defined below:

**Normal payments only:** Standard listed payments paid in exchange for completing a task. Payments are contingent on the task being approved for work quality.

**Bonus payments** in addition to normal payments: Additional payments given to workers after they have completed the task in addition to the advertised payment rate.

Q2 Opinion on payment type:

(1=Dislike, 5=Like)

| | 1 | 2 | 3 | 4 | 5 |
|---|---|---|---|---|---|
| Bonus Payments | | | ● | | |
| Normal Payments | | | ● | | |

Q3 (Optional) Please elaborate on your above response:

_______________________________________________________________

Q4 How will the payment type influence the amount of effort you put into your responses?

(1=Reduced response effort, 5=Increased response effort)

| | 1 | 2 | 3 | 4 | 5 |
|---|---|---|---|---|---|
| Bonus Payments | | | ● | | |
| Normal Payments | | | ● | | |

Q5 (Optional) Please elaborate on your above response:

_______________________________________________________________

Figure 4: Full survey (page 1).

Q6 How will the payment type influence the likelihood that you will complete the task?

(1=Much lower likelihood of completion, 5=Much higher likelihood of completion)

|  | 1 | 2 | 3 | 4 | 5 |
|---|---|---|---|---|---|
| Bonus Payments | | | ▮ | | |
| Normal Payments | | | ▮ | | |

Q7 (Optional) Please elaborate on your above response:

_____________________________________________________________

Q8 Do you have a minimum threshold for payment per unit of time that determines whether you will complete a task? (If so, please indicate dollar amount per unit time, e.g. $15 per 60 minutes)

○ No Minimum

○ Yes, I have a minimum: _____________________________________________

Q9 As a follow up for your previous response, what is the rationale behind your minimum threshold? (Select all that apply)

☐ I want my payment to be comparable to the wage I can earn elsewhere

☐ The payment level indicates a more reliable Requestor

☐ It's not worthwhile for me to complete tasks that pay less than that rate

☐ Other ______________________________________________

Figure 5: Full survey (page 2).

Q10 Are there certain types of questions that will cause you to decide to not complete a task? (Select all that apply)

- ☐ Demographic: Race
- ☐ Demographic: Gender
- ☐ Demographic: Sexual Orientation
- ☐ Demographic: Age
- ☐ Demographic: Education
- ☐ Demographic: Employment
- ☐ Demographic: Disability
- ☐ Demographic: Political stance
- ☐ Questions containing offensive content (e.g. tasks containing passages asking to identify the presence of harmful language)
- ☐ Questions concerning your mental health
- ☐ Other _______________________________________________

Q11 Under what circumstances would you respond to a question untruthfully?

- ☐ When presented with questions that feel like a privacy violation. (Please describe specific questions you've come across that apply)
  _______________________________________________
- ☐ Concerns that the question responses may be used against you. (Please describe specific questions you've come across that apply)
  _______________________________________________
- ☐ Other _______________________________________________

Q12 Are there certain types of questions that will cause you to answer untruthfully? (Select all that apply)

- ☐ Demographic: Race
- ☐ Demographic: Gender
- ☐ Demographic: Sexual Orientation
- ☐ Demographic: Age
- ☐ Demographic: Education
- ☐ Demographic: Employment
- ☐ Demographic: Disability
- ☐ Demographic: Political stance
- ☐ Questions containing offensive content (e.g. tasks containing passages asking to identify the presence of harmful language)
- ☐ Questions concerning your mental health
- ☐ Other _______________________________________________

Figure 6: Full survey (page 3).

Q13 Have you ever felt that any of the demographic questions and/or their response options do not fully capture your identity?

- [ ] Demographic: Race
- [ ] Demographic: Gender
- [ ] Demographic: Sexual Orientation
- [ ] Demographic: Age
- [ ] Demographic: Education
- [ ] Demographic: Employment
- [ ] Demographic: Disability
- [ ] Demographic: Political stance
- [ ] Other _______________________________________________

Q14 (Optional) Please elaborate on your above response:
_________________________________________________________________

Q15 If you were considering completing a survey containing potentially sensitive or harmful questions, what information would you like to be provided to you?

- ○ General content warnings (e.g. "offensive content")
- ○ Specific content warnings (e.g. "homophobia")
- ○ Mental health resources
- ○ Other (Please elaborate) _______________________________________________

Q16 In MTurk tasks, although your name is not included with the survey results, WorkerIDs that uniquely identify each responder are stored with your responses to MTurk tasks. Were you aware of this, and is this a concern for you?

- [ ] I was aware, and it is not a concern
- [ ] I was aware, and it is a concern, but it has not caused me to change my behavior when completing tasks
- [ ] I was aware, and it is a concern that may cause me to avoid answering certain questions
- [ ] I am aware, and it is a concern that may cause me to avoid working on certain tasks
- [ ] I was not aware, and it is not a concern
- [ ] I was not aware, and this knowledge will affect my responses and selection of surveys to complete in the future

Figure 7: Full survey (page 4).

Q17 Have you encountered a task that specifies that you must complete the task within a certain absolute range of response times (e.g. between 5 minutes and one hour) to be approved? If so, how has that impacted the way you complete the task?

- ○ No, I haven't seen such a task
- ○ I have seen such a task but haven't worked on it
- ○ Spent time doing other things to extend work time
- ○ Spent more time thinking about responses for each task (for minimum required)
- ○ Other (Please elaborate) _______________________________________________

- - - - - - - - - - - - - - - - - - - - - - - - - - - - - - - - - - - - - - - - - - - - - - - - - - - - - - - - - -

Q18 Have you completed a task that specifies that a certain percentile of response times (e.g. all responses in the bottom and top 5%) will be rejected? If so, how has that impacted the way you complete the task?

- ○ No, I haven't seen such a task
- ○ I have seen such a task but haven't worked on it
- ○ Spent time doing other things to extend work time
- ○ Spent more time thinking about responses for each task (for minimum required)
- ○ Other (Please elaborate) _______________________________________________

Q19 Tasks on MTurk typically list the expected amount of time needed to complete the task. What is your perception of this?

| | 1 | 2 | 3 | 4 | 5 |
|---|---|---|---|---|---|
| 1=Estimates are shorter than actual completion time 5=Estimates are longer than actual completion time | | | ● | | |

Q20 Sometimes, tasks on MTurk include a timer that limits the amount of time one can spend on a task. Please answer the following questions regarding this feature:

| | 1 | 2 | 3 | 4 | 5 |
|---|---|---|---|---|---|
| What is your general perception of this? (1=Negative, 5=Positive) | | | ● | | |
| How reasonable is the timer? (1=Time limit is too short, 5=Time limit is far above the time needed) | | | ● | | |
| How does this timer influence your response quality? (1=I put less effort, 5=I put more effort) | | | ● | | |

Figure 8: Full survey (page 5).

Q21 Many tasks on MTurk have requirements that workers must meet to qualify to complete the task. Have any of them felt unreasonable?

- ☐ Custom qualifications (if so, list an example)
  _______________________________________________
- ☐ Premium qualifications (e.g. language fluency; household income; employment)
- ☐ Masters qualification
- ☐ Hit approval rate
- ☐ Number of HITs approved

Q22 (Optional) Please elaborate on your above response:
  ______________________________________________________________

Q23 What do you think is a reasonable hit approval rate to require for a task?

- ○ Provide a percentage: ______________________________________________
- ○ Don't know

Q24 Which of the factors below would most effectively encourage you to put more effort into your responses? (Select all that apply)

- ☐ High(er) payment
- ☐ Bonus payments
- ☐ Clear/protective data sharing policies
- ☐ Difficulty of understanding task
- ☐ Readability and ease of understanding task
- ☐ Approval rate of requester
- ☐ Accurate time estimation
- ☐ Other Turkers' experience with the requester (as shared on platforms such as Reddit)

Q25 Which of the factors below would most effectively encourage you to decide to start a task? (Select all that apply)

- ☐ High(er) payment
- ☐ Bonus payments
- ☐ Clear/protective data sharing policies
- ☐ Difficulty of understanding task
- ☐ Readability and ease of understanding task
- ☐ Approval rate of requester
- ☐ Accurate time estimation
- ☐ Other Turkers' experience with the requester (as shared on platforms such as Reddit)

Figure 9: Full survey (page 6).

Q26 Have you ever had a response unfairly rejected? If so, indicate for what reasons:

- ○ Failing attention checks
- ○ Inaccuracy of response
- ○ Completing in too short a time
- ○ Completing in too long a time
- ○ Wasn't told why
- ○ Other (Please elaborate) _______________________________________________

Q27 Please rank the following aspects of a task that influence your perception of how "reasonable" and "reliable" a requestor is on MTurk (where 1 is most influential, 5 is least influential)

_______ High payment
_______ Number of previously published tasks
_______ Clarity of instructions
_______ Length of task
_______ Design and accessibility of task
_______ Other Turkers' experience with the requester (as shared on platforms such as Reddit)

Q28 Annotation tasks are tasks on MTurk that provide a textual passage (e.g. a post on Reddit) and ask the worker to evaluate it in some way. Examples of this could include: 1) highlighting the answer to a question in a short passage, 2) Given two pieces of text, selecting the more preferred/more fluent/better response, 3) indicating if there is positive or negative sentiment in the provided text. Have you ever completed an annotation task on MTurk?

- ○ Yes
- ○ No

Q29 What part of annotation tasks (as described above) do you most commonly find confusing?
_______________________________________________________

Q30 Have you ever come across questions where you were not sure about the correct or best answer to put? If so, what would have been most helpful to you in answering the question? Please elaborate on your choice with your specific experience(s). (Select all that apply)

- ☐ Context
- ☐ Purpose of question
- ☐ Platform that annotations would be used for (for annotation tasks)
- ☐ Other

Q31 (Optional) Please elaborate on your above response:
_______________________________________________________

Figure 10: Full survey (page 7).