# OpenReview forum: "Incorporating Worker Perspectives into MTurk Annotation Practices for NLP"
_EMNLP/2023/Conference — EMNLP 2023 Main_

### Official Review · Reviewer_sUkx · 2023-07-31

**Soundness:** 3

**Excitement:**

3: Ambivalent: It has merits (e.g., it reports state-of-the-art results, the idea is nice), but there are key weaknesses (e.g., it describes incremental work), and it can significantly benefit from another round of revision. However, I won't object to accepting it if my co-reviewers champion it.

**Paper Topic And Main Contributions:**

This paper conducts a critical literature review and a survey of MTurk workers to address open questions regarding best practices for fair payment, worker privacy, data quality and worker incentives.
Author conclude their findings in this paper and provide suggestions on how to improve the quality of data annotation from MTurk.

**Reasons To Accept:**

1. Manual data annotations are essential in various AI tasks. MTurk is a popular platform for annotations. This paper reveals multiple aspects of the data annotations. The authors conduct a survey and conclude the potential issues that can exist in the current MTurk workers.

2. The suggestions from this paper can provide some useful advice for the future MTurk usage.

**Reasons To Reject:**

1. Although users claim this paper is for NLP tasks, most content of this paper is general for different domains of AI and I didn't find some specific conclusion used for NLP tasks.

2. Only 207 workers completed the questionnaire in this paper, hence, the results may be less representative.

3. Not sure if this paper is suitable to be a research track paper in EMNLP. Authors conclude some phenomena from MTurk annotators and provide their own suggestions.

**Reproducibility:**

3: Could reproduce the results with some difficulty. The settings of parameters are underspecified or subjectively determined; the training/evaluation data are not widely available.

**Reviewer Confidence:**

2: Willing to defend my evaluation, but it is fairly likely that I missed some details, didn't understand some central points, or can't be sure about the novelty of the work.

---

> ### Author Rebuttal · Authors · 2023-08-28
>
> Thank you very much for taking the time to provide helpful feedback on our paper. We will integrate your comments into the camera-ready version.
>
> > “Although users claim this paper is for NLP tasks, most content of this paper is general for different domains of AI and I didn't find some specific conclusion used for NLP tasks.”
>
> Although much of the content would be useful to any AI researcher doing crowdsourcing work, crowdsourcing is particularly critical to NLP (as reviewer PCxw noted). The percent of NLP papers involving crowdsourcing grew steadily from 2017 to 2020, with 12% of accepted papers in major conferences (ACL, EMNLP, and NAACL) in 2020 explicitly describing crowdsourcing as part of their research methods (Shmueli et al., 2021). In addition, we include several sections that focus on issues specific to text data (e.g. 3.1 and 3.2 on clarity in annotation tasks, and 6.2 on annotators’ use of LLMs for writing text assumed to be human-authored). We came at this from the perspective of NLP researchers and did not focus on investigating issues that might be relevant to other domains, such as computer vision, but are glad that the findings might be more widely applicable. We can expand on the sections that are especially specific to NLP for the camera-ready version.
>
> > “Only 207 workers completed the questionnaire in this paper, hence, the results may be less representative.”
>
> We would argue that 207 workers is a reasonable size for a survey (compare 80 respondents in Kim et al., 2021 and 435 respondents in Xia et al., 2017). In addition, because our results include manual analysis of qualitative, open-ended feedback from survey responses, we prioritized detail of analysis over volume of responses. However, we will note the potential non-representativeness of the sample as a limitation.
>
> > “Not sure if this paper is suitable to be a research track paper in EMNLP. Authors conclude some phenomena from MTurk annotators and provide their own suggestions.”
>
>
> The ACL’23 Peer Review Policies note that “NLP is an interdisciplinary field, relying on many kinds of contributions: models, resource, survey, data/linguistic/social analysis, position, and theory” (Section 4.2). Much of NLP research is heavily dependent on crowdsourced data collection. Given that crowdsourcing is both fundamental to NLP and one of the aspects most vulnerable to errors, our paper contributes to the research community by (1) providing an overview of best practices that are often spread by word-of-mouth, with no centralized resource to aid NLP researchers collecting crowdsourced data; (2) evaluating public MTurk worker comments and surveying workers, which provides novel findings about flaws of current accepted practices; and (3) providing recommendations to help NLP researchers. We argue that our results, which are grounded in the research literature and our survey findings, can directly help future NLP researchers to collect crowdsourced data with improved worker recruitment, labor practices, and data quality.
>
> References
>
> Jane Paik Kim, Katie Ryan, Tenzin Tsungmey, Max Kasun, Willa A. Roberts, Laura B. Dunn, and Laura Weiss Roberts. 2021. Perceived protectiveness of research safeguards and influences on willingness to participate in research: A novel mturk pilot study. Journal of Psychiatric Research, 138:200–206.
>
> Boaz Shmueli, Jan Fell, Soumya Ray, and Lun-Wei Ku. 2021. Beyond Fair Pay: Ethical Implications of NLP Crowdsourcing. In Proceedings of the 2021 Conference of the North American Chapter of the Association for Computational Linguistics: Human Language Technologies, pages 3758–3769, Online. Association for Computational Linguistics.
>
> Huichuan Xia, Yang Wang, Yun Huang, and Anuj Shah. 2017. "Our privacy needs to be protected at all costs": Crowd workers’ privacy experiences on Amazon Mechanical Turk. Proc. ACM Hum.-Comput. Interact., 1(CSCW).

---

### Official Review · Reviewer_NbBA · 2023-08-05

**Soundness:** 3

**Excitement:**

3: Ambivalent: It has merits (e.g., it reports state-of-the-art results, the idea is nice), but there are key weaknesses (e.g., it describes incremental work), and it can significantly benefit from another round of revision. However, I won't object to accepting it if my co-reviewers champion it.

**Paper Topic And Main Contributions:**

The paper argues for including perspective of MTurk workers for improving collected data quality and improving worker rights. They prescribe best practices for data collection using MTurk, considering various aspects including fair pay, privacy concerns and sensitive content.

The paper is very well written (as someone who is not familiar with the relevant literature, I had no issues following the points in the paper)

**Reasons To Accept:**

The paper presents strong, logical and actionable steps for better data collection. The paper considers various aspects of the data collection, instead of being limited to a single aspect.

The paper is very well written (as someone who is not familiar with the relevant literature, I had no issues following the points in the paper)

**Reasons To Reject:**

Not being familiar with the literature, I don’t see any apparent reasons to reject the paper.

**Reproducibility:**

N/A: Doesn't apply, since the paper does not include empirical results.

**Reviewer Confidence:**

1: Not my area, or paper was hard for me to understand. My evaluation is just an educated guess.

---

> ### Author Rebuttal · Authors · 2023-08-28
>
> Thank you very much for taking the time to provide feedback on our paper. If you have any specific questions or concerns, we would be happy to address them.

---

### Official Review · Reviewer_PCxw · 2023-08-05

**Soundness:** 5

**Excitement:**

4: Strong: This paper deepens the understanding of some phenomenon or lowers the barriers to an existing research direction.

**Missing References:**

- Crowdworker Economics in the Gig Economy, Jaques and Kristensson 2019. (https://dl.acm.org/doi/10.1145/3290605.3300621)

**Paper Topic And Main Contributions:**

The authors take a three-fold approach to investigate crowdworker perspectives on MTurk: They (i) review related work, (ii) evaluate public comments (from Reddit), and (iii) conduct a large scale survey on MTurk.
The results are grouped along five guiding themes: Task clarity, payment, privacy, response quality, and sensitive content.
Importantly, the paper consolidates information that is partly already known by experienced practitioners (e.g., that using the Master qualification is unfair to new workers) but not available as a united resource yet.
In addition, it reports interesting findings (e.g., regarding not paying too much) that I consider to be equally surprising and helpful.
Having had this paper a couple years ago would have saved me a lot of trouble and money, and I would recommend the paper to all my students working on crowdsourcing on MTurk.

**Reasons To Accept:**

- The authors consolidate many useful practices that are know by word of mouth, but have, to the best of my knowledge, not been consolidated yet.
- The authors report very important insights that, after conducting crowdsourcing studies for various years, a new to me (e.g., regarding paying too much).
- The authors conduct a large-scale survey and report all survey details.
- The paper is well written and follows a clear and comprehensive structure by grouping literature survey results and questionnaire results together per aspect.
- Including Reddit comments as a complementing source to the literature review and the questionnaire is a good idea and makes the paper more tangible.

**Reasons To Reject:**

I do not see a real reason to reject this paper!
- Minor: The missing direct direct link to NLP can be criticised, but given the important role of crowdsourcing in NLP, I argue that this paper is suitable for the conference anyways.
- Minor: The paper only focuses on MTurk, replicating the study on alternative platforms would be interesting, however, this cannot be held against the authors as they already conducted a large scale study which, I think, is preferable to a series of smaller studies.

**Reproducibility:**

5: Could easily reproduce the results.

**Reviewer Confidence:**

4: Quite sure. I tried to check the important points carefully. It's unlikely, though conceivable, that I missed something that should affect my ratings.

---

> ### Author Rebuttal · Authors · 2023-08-28
>
> Thank you very much for taking the time to provide helpful feedback on our paper. We will integrate your comments into the camera-ready version:
> - We will mention the importance of replicating the study on alternative platforms as future work.
> - We will add a reference to Jaques and Kristensson, 2019 in the camera-ready version.

---

### Meta-Review · Area_Chair_jPKJ · 2023-09-18

**Recommendation:** 5

**Metareview:**

This manuscript seeks to deal with the confusion that arises from the terms (and research on) values, morals, and ethics. The manuscript surveys prior literature from outside of NLP on these concepts, then surveys NLP literature, and suggests future avenues for NLP.

In general, the reviewers are positive towards this manuscript and particularly appreciate:

1. That there is confusion of the terms and this manuscript may provide a starting point to address such confusion
2. The manuscript provides a comprehensive report on how the terms have been used
3. The manuscript provides quantitive support for their recommendations and findings
4. The manuscript highlights flaws in terminology in prior work

However, the reviewers had the following contention:

1. The manuscript would do well to consider additional related terms such as norms, beliefs, customs, behaviors, and ideologies.
2. It can be hard to gain a sense of the bigger picture from how results are presented
3. The selection of papers is narrow and does not take into consideration topics where the terms in question are implicit.
4. It is unclear how to implement recommendations for common terminology in future work

Wrt. 1 both authors and reviewers agree that this manuscript can serve as a starting point for such considerations. To this effect, authors will reframe writing to make the bigger picture (2) clearer. While the authors do not address 3, the scope of addressing this is incredibly large and is in my opinion more appropriate for future work. Finally, the authors correctly identify that NLP researchers creating vocabularies may lead to further inaccuracies and limit opportunities for interdisciplinary collaboration.

5193

This manuscript presents a study on practices and experiences of annotation on amazon mechanical turk. Towards the goal of understanding crowdwork, they review prior work, public discussions on reddit, and conduct a survey on MTurk. As a result they provide several recommendations based on their findings.

Reviewers are generally positive towards this manuscript, particularly as it provides evidence-based recommendations for best-practices for annotation.

Of the concerns highlighted, only the concern that the manuscript applies to more fields than NLP is reasonable. While the reviewer is correct that best practices for annotation work is not only applicable to NLP, work in NLP very often requires human labelling and the paper is therefore also very applicable to the ACL community.

A slight concern that I have: The book Ghost Work by Suri and Grey is missing from the references, which would further support and contextualize the claims and findings. Moreover, Turkopticon, the organization that has been working on crowd worker’s rights seems to have been neglected, and the findings from their work and their advocacy should be included in the final manuscript, should it be accepted.

---

### Decision · Program_Chairs · 2023-10-07

**Decision:**

Accept-Main

**Comment:**

This manuscript seeks to deal with the confusion that arises from the terms (and research on) values, morals, and ethics. The manuscript surveys prior literature from outside of NLP on these concepts, then surveys NLP literature, and suggests future avenues for NLP.

In general, the reviewers are positive towards this manuscript and particularly appreciate:

1. That there is confusion of the terms and this manuscript may provide a starting point to address such confusion
2. The manuscript provides a comprehensive report on how the terms have been used
3. The manuscript provides quantitive support for their recommendations and findings
4. The manuscript highlights flaws in terminology in prior work

However, the reviewers had the following contention:

1. The manuscript would do well to consider additional related terms such as norms, beliefs, customs, behaviors, and ideologies.
2. It can be hard to gain a sense of the bigger picture from how results are presented
3. The selection of papers is narrow and does not take into consideration topics where the terms in question are implicit.
4. It is unclear how to implement recommendations for common terminology in future work

Wrt. 1 both authors and reviewers agree that this manuscript can serve as a starting point for such considerations. To this effect, authors will reframe writing to make the bigger picture (2) clearer. While the authors do not address 3, the scope of addressing this is incredibly large and is in my opinion more appropriate for future work. Finally, the authors correctly identify that NLP researchers creating vocabularies may lead to further inaccuracies and limit opportunities for interdisciplinary collaboration.

5193

This manuscript presents a study on practices and experiences of annotation on amazon mechanical turk. Towards the goal of understanding crowdwork, they review prior work, public discussions on reddit, and conduct a survey on MTurk. As a result they provide several recommendations based on their findings.

Reviewers are generally positive towards this manuscript, particularly as it provides evidence-based recommendations for best-practices for annotation.

Of the concerns highlighted, only the concern that the manuscript applies to more fields than NLP is reasonable. While the reviewer is correct that best practices for annotation work is not only applicable to NLP, work in NLP very often requires human labelling and the paper is therefore also very applicable to the ACL community.

A slight concern that I have: The book Ghost Work by Suri and Grey is missing from the references, which would further support and contextualize the claims and findings. Moreover, Turkopticon, the organization that has been working on crowd worker’s rights seems to have been neglected, and the findings from their work and their advocacy should be included in the final manuscript, should it be accepted.